# Determination of Hazard Due to Debris Flows

**Ricardo A. Bocanegra [1],\*, Carlos A. Ramírez [2], Elkin de J. Salcedo [3] and María Paula Lorza Villegas [4]**

[1] Faculty of Engineering, Universidad Santiago de Cali, Cali 760035, Colombia
[2] Faculty of Engineering, Universidad del Valle, Cali 760035, Colombia; carlos.ramirez@correounivalle.edu.co
[3] Department of Geography, Faculty of Humanities, Universidad del Valle, Cali 760035, Colombia; elkin.salcedo@correounivalle.edu.co
[4] Department for Water Resources & Flood Risk Management, Wupper Association (Wupperverband), 42001 Wuppertal, Germany; mariapaulalorzavillegas@gmail.com
\* Correspondence: ricardo.bocanegra00@usc.edu.co; Tel.: +57-315-510-84-91

**Abstract:** Debris flows have generated major disasters worldwide due to their great destructive capacity, which is associated with their high energy levels and short response times. To achieve adequate risk management of these events, it is necessary to define as accurately as possible the different hazard levels to which the territory is exposed. This article develops a new methodology to estimate this hazard based on the hydrodynamic characteristics of the flow and the granulometry of the sediments that can be mobilized by the flow. The hydrodynamic characteristics of the flow are determined via mathematical modeling that considers the rheology of non-Newtonian flows and the different volumes of sediments that could be transported during events corresponding to different return periods. The proposed methodology was implemented in the Jamundí River basin (Colombia). The results obtained indicate that in the upper part of this basin, there is a low hazard level, while in the lower part of the basin, approximately 15% of the affected territory has a medium hazard level, and the remaining 85% has a low hazard level. The methodology developed is simple to implement but technically rigorous since it considers all relevant aspects in the generation of debris flows.

**Keywords:** debris flow; hazard zoning; mathematical modeling; intensity of debris flow; debris size

## 1. Introduction

Debris flows have great destructive capacity and are associated with terrible social and economic consequences worldwide [1–4]. Due to their high energy levels and short response times, these events have caused the loss of a large number of human lives and substantial material losses [5,6]. According to [7], annually, these events generate, on average, approximately 1200 deaths. In the European Alps between 1987 and 2012, debris flows caused the death of more than 200 people and losses of more than 5 billion euros [8]. In Colombia, between 1921 and 2018, the occurrence of 1358 events was reported, which caused the death of 3318 people, affected 1,264,705 people, destroyed 13,698 houses, and affected another 23,694 houses [9]. In Venezuela, heavy rains between 14 and 16 December 1999 generated debris flows that caused about 30,000 deaths and losses estimated at $1.79 billion [10]. On 3 July 2021, large debris flows in Atami, Japan, killed 26 people and damaged 128 houses [11].

Additionally, due to the effects of climate change and the increased exposure of people and infrastructure, the consequences of these events are expected to be increasingly critical [12,13]. However, despite the fact that debris flows represent a great hazard to humans and infrastructure [14] and that the transported sediments have a strong influence on the characteristics of these flows [2], to date, there are few studies available for the determination of this hazard that consider the flow rheology and all the volumes of sediments that could be transported, as for example, those from the potential erosion of the channels. Most available methodologies are based on morphometric parameters [15,16] and in empirical equations [17–19].

Based on the value of a hazard index H, ref. [20] classified the hazard due to debris flows in Wenchuan County into five categories: very low, low, moderate, high, and very high. The hazard intensity increases as the value of the H index increases, which fluctuates between 0 and 1 and is calculated from the magnitude of seven parameters: maximum depositional volumes of a debris flow ($10^3$ m$^3$), frequency of debris flow occurrence scaled to the times per century (%), drainage basin area (km$^2$), main channel length (km), drainage basin relief (km), drainage density (km/km$^2$), and active main channel proportion (%). To calculate the H index, a sum is made of the results obtained by assigning each of these seven parameters a value between 0 and 1 via transformation functions.

Ref. [21] determined the hazard in Du Jiangyan City, China, by combining land uses and depths reached by debris flows during events corresponding to return periods of 5, 10, 20, 20, 50, and 100 years. This methodology considers three land uses (construction, transportation, and forest) and four different depths (greater than 4 m, between 2 and 4 m, between 1 and 2 m, and less than 1 m), resulting in twelve combinations from which the hazard classification is made in three categories: high, moderate, and low. The authors calculated the hazard for each return period analyzed.

Ref. [22] proposed a methodology for hazard zoning due to debris flows based on the evaluation of previous events, the identification of the main geomorphological factors, the determination of the magnitude of future events via the implementation of empirical equations, and the identification of elements at risk including infrastructure and populations. This methodology allows the mapping of areas subject to critical hazard levels.

Among the studies that somehow take into account the flow rheology, we find research presented by [14], who proposed the determination of the debris flow hazard in areas influenced by hydroclimatic events via the implementation of four basic steps: initially, logistic regression is performed in order to determine the factors that have a greater influence on the generation of debris flow; next, a numerical simulation of previous events is carried out using a two-dimensional mathematical model that considers the rheology of a debris flow; subsequently, an analysis of the precipitation prior to the occurrence of the flood is performed; and, finally, based on the results of the previous analyses, a hazard classification matrix is generated from which it is possible to carry out zoning of the territory.

Ref. [23] conducted a classification of the debris flow hazard based on event intensity and return period. In this methodology, using a two-dimensional model that takes into account flow rheology, a numerical simulation of events corresponding to return periods of 50 and 100 years is carried out. Subsequently, using the results of the numerical simulation, the degree of impact of the event is classified into high, medium, or low according to the values obtained from the depth and the product of the depth by the flow velocity. Finally, by integrating the degree of impact of the event with the frequency of occurrence of the events analyzed, the hazard is classified as high, medium, and low. This methodology was applied to assess the hazard due to debris flow in Wudu District, Northwest China.

This paper proposes a new methodology for hazard zoning due to debris flows. In this methodology, the hazard is established based on a flow intensity index, the probability of occurrence of the events, and the characteristics of the sediments that can be transported by the flow. The flow intensity index is calculated from the flow velocities and depths, which are obtained via hydrodynamic modeling that considers the rheology of non-Newtonian flows.

This article first briefly describes the phenomenon of debris flows. Next, the methodology proposed for the determination of the hazard by debris flows based on the hydrodynamic characteristics of the flow and the granulometry of the sediments is described. Subsequently, the implementation of the methodology in the Jamundí River basin (Colombia) is presented. Finally, the main conclusions obtained from the development and implementation of the proposed methodology are presented.

## 2. Intensity of Debris Flows

Debris flows correspond to rapid flows of water and sediments mixed in different proportions that transit through channels with steep slopes, generating a short response time [5]. The sediments transported during this type of event, whose concentration varies between 40 and 80% [24], can come from the riverbed itself [25,26] and from the surrounding slopes when pluvial erosion or destabilization of slopes occurs [27,28]. The transported material is deposited when the flow reaches low slopes [29,30].

Similarly, in the marine environment, a debris flow corresponds to a laminar flow with a sediment concentration that varies between 25 and 100% and presents a plastic rheology [31].

On the land surface, debris flows can be triggered by precipitation, snowmelt, changes in water level (water table), stream erosion, earthquakes, volcanic activity, disturbance by human activities, or any combination of these factors [32]. On most occasions, these types of events are triggered by heavy rainfall [33,34].

According to [35], the destructive capacity of debris flows is associated with three types of forces: (i) the hydrodynamic force, which corresponds to a combination of the frontal impact of the flow, the dredging effect on the sides of the structure, etc.; (ii) the hydrostatic force; and (iii) the collisional force due to the debris transported by the flow. The magnitude of these forces is a function of the peak discharge, depth, flow velocity, and the volume, concentration, and granulometric distribution of the transported sediments.

*Debris Flow Intensity Index*

In order to classify the hazard generated by debris flows, it is necessary to define a criterion that allows establishing the intensity of the events, and that can be associated with different levels of damage observed in events that occurred previously. To establish this criterion, several authors have proposed various expressions, some of them based on empirical and semi-empirical equations and others based on the kinetic energy of classical mechanics for a rigid body [36].

In this paper, the Debris Flow Intensity Index, $I_{DF}$, proposed by [36], was selected since it can represent the forces that generate damage and, therefore, can be correlated with infrastructure damage. This index is calculated according to the following expression:

$$I_{DF} = d_{max} \times V_{max}^2 \tag{1}$$

where $I_{DF}$ is the Debris Flow Intensity Index (m$^3$/s$^2$), $d_{max}$ is the maximum flow depth (m), and $V_{max}$ is the maximum flow velocity (m/s).

## 3. Materials and Methods

Hazard by debris flow corresponds to the probability of the occurrence of a debris flow with the potential to generate damage in a given site during a certain period of time. The probability of occurrence can be expressed as the frequency of occurrence, which is indicated via the return period.

Hazard due to debris flows is obtained by combining the probability of occurrence of this type of event with indicators of the magnitude of such events. A similar approach in some aspects is followed by [37].

Considering that the process of mathematical modeling of watersheds prone to experiencing debris flows is highly demanding in terms of information and computational time, the methodology proposed in this paper is composed of two phases, which are described below.

### 3.1. Phase I: Determination of the Hazard Due to Debris Flows in Rural Areas

Given that the dimensions of the areas susceptible to the phenomenon of debris flows in rural areas can be very large, the simulation process could become very time-consuming. Because of this, the first step is to perform a mathematical modeling of the areas exposed to

debris flows using a digital elevation model, DEM, at a scale of 1:25,000 or smaller. The modeling of events must be carried out to cover a wide range of events, for which it is suggested to model the following return periods: 2.33, 5, 10, 25, 50, 75, 100, 200, 300, 400, and 500 years. These return periods refer to the recurrence interval of the rainfall events that have occurred in the basin, which should be recorded in climatological, limnimetric, or limnigraphic stations. The discharges associated with these precipitation events are determined via hydrological studies.

For the numerical simulation, a two-dimensional mathematical model averaged in depth should be used that considers the rheology of non-Newtonian flows and allows the incorporation of sediments from slope erosion, landslides, and rockfalls. Some of the available models that can be used for this purpose are FLO2D, RAMMS, RIVERFLOW2D, HEC-RAS, AND FLATMODEL. These models use different flow resistance expressions: some of them assume single-phase flow, and others assume two-phase flow; some are free, while others are commercial; and some have a user-friendly interface. For the selection of the model to be used, the amount of information available for the area to be modeled and the assumptions and limitations of each one of them should be taken into account.

The input information to a model of these characteristics is constituted by the DEM at scale 1:25,000 or smaller, the hydrographs of debris flows corresponding to different return periods, the volumes of sediments associated with these events, and the rheological characteristics of the flow.

The results of the mathematical modeling correspond to the depths and velocities of flow in each of the pixels through which the territory has been represented. Using the maximum depths and flow velocities, the $I_{DF}$ is calculated in each pixel by means of Equation (1) for all the return periods analyzed.

Since a different $I_{DF}$ is obtained for each return period, in order to obtain a single value in each pixel, the following equation must be applied, which is similar to that of the mathematical expectation, with the difference that exceedance probabilities are used instead of occurrence probabilities.

$$I_{DF_{Pj}} = \left[ \sum_{i=1}^{n-1} I_{DF_{Ti}} \left( \frac{1}{T_i} - \frac{1}{T_{i+1}} \right) \right] + I_{DF_{Tn}} \left( \frac{1}{T_n} \right) \tag{2}$$

where $Pj$ is the pixel $j$; $I_{DF_{Pj}}$ is the combined Debris Flow Intensity Index of pixel $j$; $T_i$ is the return period $i$; $i$ is the number of the order of return period, which varied from 1 to $T_{min}$ up to n for $T_{max}$; $T_{min}$ is the lowest return period from which the $I_{DF}$ is calculated; $T_{max}$ is the longest return period analyzed; and $I_{DF_{Ti}}$ is the Debris Flow Intensity Index corresponding to return period $i$.

The $T_{min}$ value can correspond to the return period from which the flood begins or to the return period from which debris flows begin to have a considerable impact on the territory. The selection of the criterion to be applied will depend on the quantity and quality of information available and on expert judgment.

Since debris flows with a relatively low return period, have a stronger effect on the value of the combined $I_{DF}$ than less frequent events, the value of $T_{min}$ must be obtained with the greatest possible certainty in order to avoid overestimating or underestimating the value of this parameter [38].

In addition to depth and flow velocity, the intensity of debris flows is determined by other factors, including slope geometry and local changes in their morphology, antecedent hydrological conditions, and size of the sediment [39]. In this methodology, the size of the sediment was used as an additional variable to establish the magnitude of the event because several authors [40–42] have pointed out that the capacity of debris flows to generate damage is directly related to the size of the largest blocks transported by the flow. However, considering that the size of the largest block transported by the flow, $D_{max}$, can lead to an overestimation of the impact of the event since it could not be representative of the material transported and the precise estimation of the $D_{max}$ that could be transported

is complex, it is recommended to use the $D_{90}$ as a representative size of the blocks that could be mobilized, considering that the $D_{90}$ corresponds to a diameter such that 90% of the diameters of the transported sediments are smaller than this value.

There are several classifications of sediment size, some of which are presented in [43]. In the present methodology, the classification presented by [41] was adopted, according to which sediments transported in a debris flow can be classified as fine particles, coarse particles, and boulders. Fine particles are those with a diameter of less than 1 cm, coarse particles have diameters that fluctuate between 1 and 50 cm, and boulders have diameters greater than 50 cm. Based on this classification, three ranges of sediment sizes have been established in this methodology for hazard estimation: the first range corresponds to particles with a diameter of less than 50 cm, which includes sediments of relatively small size; the second range includes particles whose diameter fluctuates between 50 cm and 1.0 m, which are considered to be of medium and large size; and the third range corresponds to particles of a diameter greater than 1 m which are considered as quite large. A number of 1 m was established as the separation diameter between medium and large sediments with quite large sediments because several studies [44,45] have shown that boulders with this diameter can have a great impact force, which implies a high destructive power.

To classify the hazard, the variables indicating the magnitude of the debris flow are integrated, as shown in Table 1. The results obtained by applying these criteria to each pixel will finally allow us to obtain the hazard zoning map by debris flows.

**Table 1.** Classification of the hazard due to debris flows according to the Debris Flow Intensity Index, $I_{DF}$, and the $D_{90}$ of the sediments that could be mobilized.

| $D_{90}$ (m) | Hazard Due to Debris Flows | | |
|:---:|:---:|:---:|:---:|
| | Combined Debris Flow Intensity Index $I_{DF}$ (m³/s²) | | |
| | **0–1** | **1–50** | **>50** |
| 0–0.5 | Low | Medium | Medium |
| 0.5–1.0 | Medium | High | High |
| >1.0 m | Improbable | High | High |

*3.2. Phase II: Determination of Hazard in Urban Areas, Urban Expansion Areas, and Areas Classified as High and Medium Hazards in Rural Areas*

Since a debris flow can have disastrous consequences in urban sectors and in rural sectors classified as medium and high hazards, a more detailed analysis must be carried out for these areas to establish more precisely the hazard to which the territory is exposed.

The first step in determining the hazard in these areas consists of the hydrodynamic simulation of debris flows corresponding to different return periods. It is suggested to model the events corresponding to the same return periods used in the modeling of the rural area, that is, 2.33, 5, 10, 25, 50, 75, 100, 200, 300, 400, and 500 years. Since detailed information is required at this stage, a DEM of a higher resolution than that used in rural areas, preferably at a scale of 1:2000, should be used for this purpose.

A two-dimensional mathematical model must be used that includes the rheology of non-Newtonian flows and that, in addition to considering sediments from slope erosion, landslides, and rockfalls, has the capacity to represent the dragging of sediments from the scour of the river channel, which can represent a significant fraction of the total sediments transported, changing the behavior of debris flows [46]. These models usually have a higher computational cost and require more detailed information related to the properties of the flow, the topographic characteristics of the channel and the basin, and the sediments transported during the occurrence of the phenomenon. There are single-phase, two-phase, and three-phase models that meet these requirements. The selection of the type of model to use is a function of the computational cost, the amount of rheological information available,

the assumptions and limitations of the models, and the skill of the modeler. Among the available models are IRIC, R.AVAFLOW, D-CLAW, RAMMS, and RIVERFLOW2D.

The input information to these models consists of the DEM at scale 1:2000, the rheological characteristics of the flow, the hydrographs of discharges corresponding to different return periods and the estimated volumes of sediments from the scour of the channel, the surface erosion of the basin, landslides, and rockfalls for the different return periods considered.

Given that the numerical simulation of debris flows is carried out using high-quality territorial and flow information and that the models used allow considering the temporal distribution of sediments that are incorporated into the flow, as well as the dragging of sediments from the riverbed, it is considered that the results obtained provide an adequate representation of the physics of debris flows, so it is not necessary to involve flow intensity variables additional to the $I_{DF}$ (such as sediment particle size) to perform a hazard classification [37].

For each of the return periods analyzed, an $I_{DF}$ value is obtained, which increases as the frequency of occurrence of the events decreases. Using this information, a hazard curve is constructed for each pixel, taking the $I_{DF}$ values as the ordinate axis and the corresponding return periods as the abscissa axis. For each pixel involved in the flow, a curve similar to the one presented in Figure 1 is obtained. If, in some pixels, the number of $I_{DF}$ values is insufficient to plot a curve, it is suggested to carry out additional return period modeling in order to complement this information.

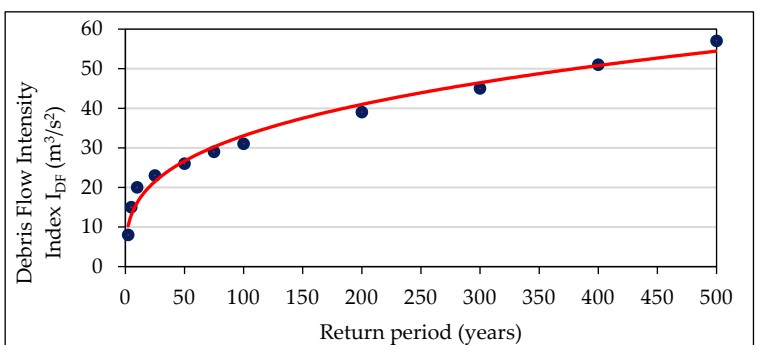

**Figure 1.** Typical hazard curve that relates $I_{DF}$ to their corresponding return periods.

To carry out the zoning of the territory, the impacts of high-frequency events and then the impacts of low-frequency events are analyzed independently, and finally, the results obtained are integrated into a single map.

For events of high-frequency occurrence, their capacity to generate considerable structural damage is evaluated, which occurs, according to the results presented by [36], when the $I_{DF}$ is greater than 5 m³/s². In the present methodology, high-frequency events were defined as those corresponding to return periods of 30 and 100 years. The value of 30 years was adopted because, in some countries, such as Colombia, it is the return period used for the design of works to protect agricultural areas against floods, and the period of 100 years was adopted because, in some countries, such as Spain, Mexico, and Colombia, it is a reference value for the determination of flood hazard and risk.

The hazard classification by debris flows of high frequency of occurrence is carried out according to the information recorded in Table 2. The return period corresponding to the $I_{DF}$ of 5 m³/s² must be interpolated from the hazard curve plotted in each pixel.

**Table 2.** Hazard classification by debris flows of high frequency of occurrence based on the $I_{DF}$ and the return period.

| Range of the Return Period Corresponding to the $I_{DF} = 5$ m$^3$/s$^2$ (Years) | Hazard Classification |
|:---:|:---:|
| $\leq$30 | High |
| $30 < T < 100$ | Medium |
| $\geq$100 | Low |

For events of low frequency of occurrence, their capacity to generate severe structural damage is evaluated. Based on the work presented by [36], it is possible to infer that this situation occurs when the $I_{DF}$ takes values higher than 25 m$^3$/s$^2$. The event corresponding to the 500-year return period was established as an event of low frequency of occurrence because it is taken as a reference in some countries, such as Spain and Colombia, for the determination of the flood hazard and risk.

For the classification of the hazard due to debris flows with a low frequency of occurrence, the $I_{DF}$ value corresponding to the 500-year return period must be determined in each pixel by means of the hazard curve, and then the hazard level must be established according to the classification presented in Table 3.

**Table 3.** Hazard classification due to debris flows with low frequency of occurrence based on the $I_{DF}$ of the event with return period of 500 years.

| Range of the $I_{DF}$ Corresponding to the Event with a Return Period of 500 Years (m$^3$/s$^2$) | Hazard Classification |
|:---:|:---:|
| <1 | Low |
| 1–25 | Medium |
| >25 | High |

Finally, the hazard zoning map by debris flows is established by integrating the two classifications obtained previously (for the events of high and low frequency of occurrence), assigning to each pixel the higher of the two hazard levels obtained in them.

Figure 2 shows a flow chart in which the procedure to be followed to determine the hazard due to debris flows is presented schematically.

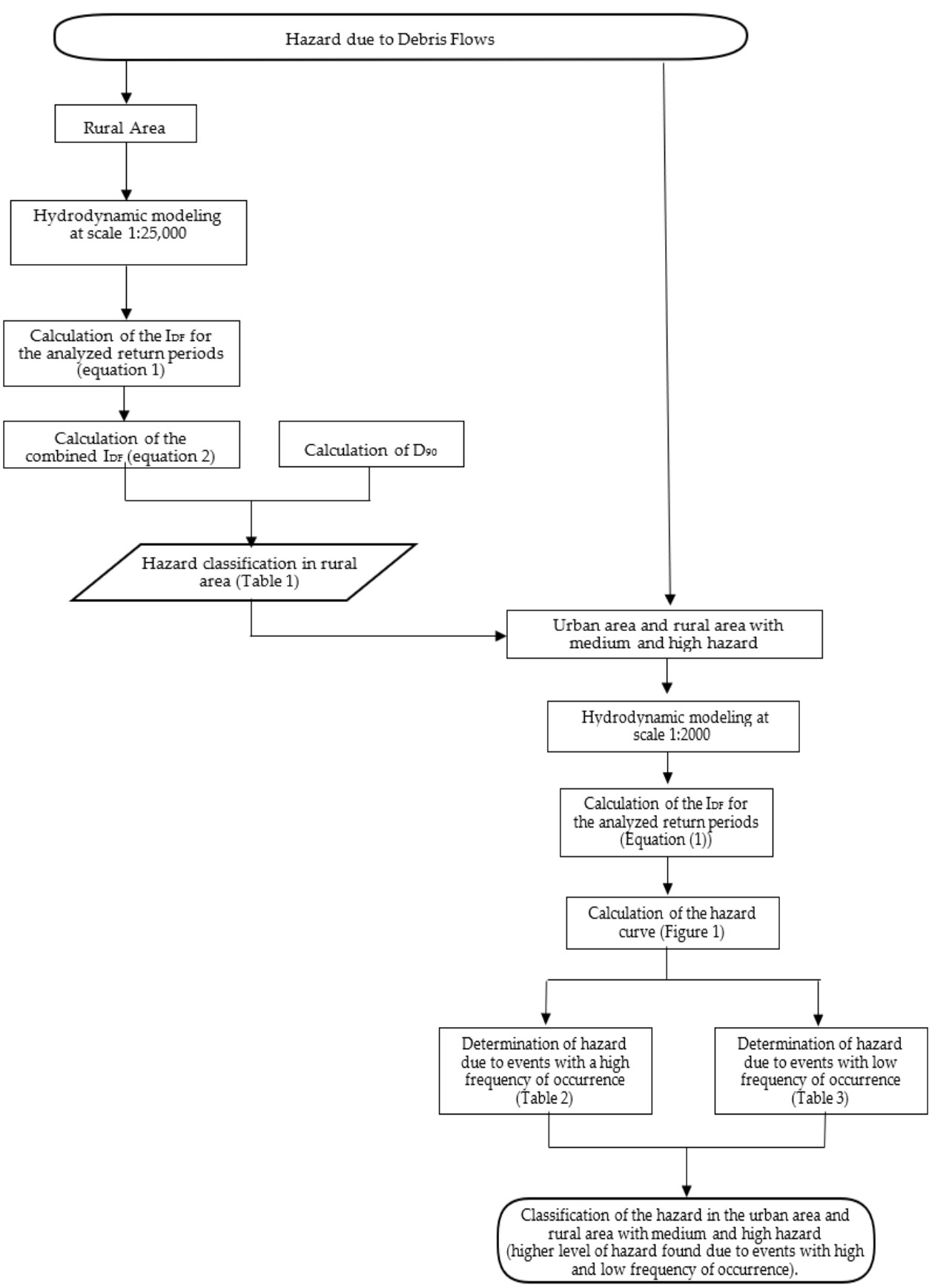

**Figure 2.** Flow chart of the procedure to be implemented to determine the hazard due to debris flows.

## 4. Application to a Case Study

In order to establish its applicability, the proposed methodology was used to determine the hazard due to debris flows in the Jamundí River basin, which is located in Colombia (Figure 3).

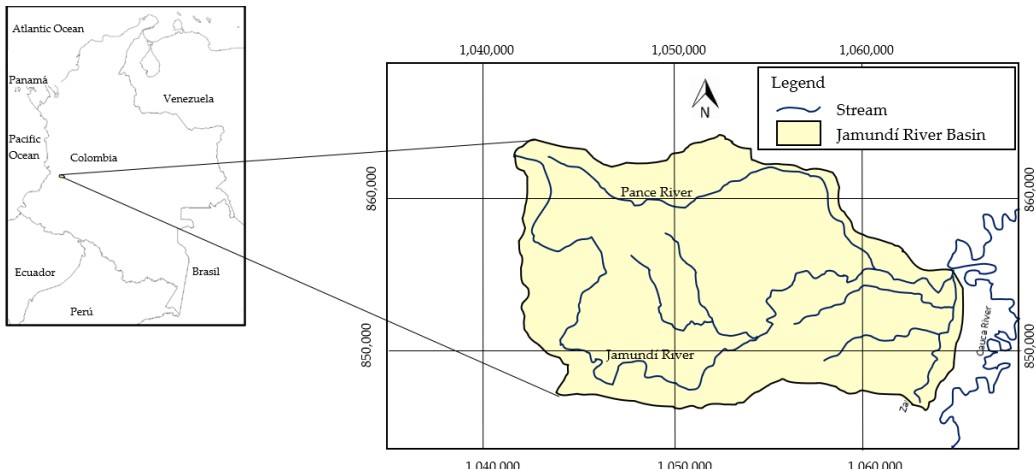

**Figure 3.** Location of the Jamundí River basin.

### 4.1. Characterization of the Study Area

The Jamundí River has a length of 41.2 km; it is born in the Colombian western mountain range at an altitude of 3900 masl and flows into the Cauca River at an altitude of 950 masl. Its drainage basin has an area of 164 km$^2$ and an average slope of 28%. It has an average discharge of 10.90 m$^3$/s and a bimodal rainfall regime with two rainy periods (March to May and October to December) and two periods of moderate rainfall (January to February and June to September). The average annual precipitation of the basin fluctuates between approximately 3500 mm in the upper part and approximately 2200 mm in the lower part. In its basin are the municipality of Jamundí and the southern part of the municipality of Cali [47]. The municipality of Jamundí, which could be seriously affected by a debris flow of the Jamundí River, has a population of 131,000 people, of which approximately 80% live in the urban area [48].

In terms of local geology, the study area is made up of a set of Quaternary deposits, mainly of alluvial and fluvial-torrential origin, associated with the Jamundí and Jordán rivers, which form bars, flood plains, terraces, and sub-recent and ancient alluvial fans. The ancient alluvial fans, on which a sector of the urban center of the municipality of Jamundí rests, are part of a group of coalescent fans deposited on the tectonic depression of the Cauca River valley, which extends toward the east from the eastern foothills of the western mountain range.

In the study area, there are 22 superficial lithological units comprising 16 transported soils, 5 anthropic soils, and 1 rock unit. Transported soils make up 93.4% of the study area and correspond to materials associated with fluvial processes, especially floodplain soils, diversion channels, and recent fluvial-torrential terraces. Anthropic soils are heterogeneous materials that result from human activities, making up 6.5% of the study area. They include materials such as road and housing construction fills, fill soils, and dike fills. The rock unit is relatively small, underlies Quaternary alluvial deposits, has a sedimentary origin, and is moderate weathering.

### 4.2. Input Information to the Mathematical Models

For the determination of the hazard, mathematical modeling of debris flows corresponding to different return periods must be carried out using two different types of mathematical models, in which different input information is introduced. This information, corresponding to the DEM, hydrological information, sedimentological information, and

rheological characteristics of the flow, was provided by the Universidad del Valle and the Colombian Geological Service [49].

### 4.2.1. Digital Terrain Elevation Models DEM

Two DEM of the study area were used, one at a scale of 1:25,000 and the other at a scale of 1:2000. These DEM were generated using cartographic information provided by the Colombian Geological Service; information taken with LIDAR technology by the CVC (regional environmental authority), which is available with a pixel size of 0.5, 1.0, and 2.5 m and centimeter vertical accuracy; information taken from the ALOS PALSAR satellite, which is available for the upper part of the basin with a pixel size of 12.5 m and submeter accuracy; high-precision GPS points located directly in the field; and, topobatimetric surveys of several channels in the Jamundí River basin. Figure 4 shows the 1:2000 scale DEM used in the modeling.

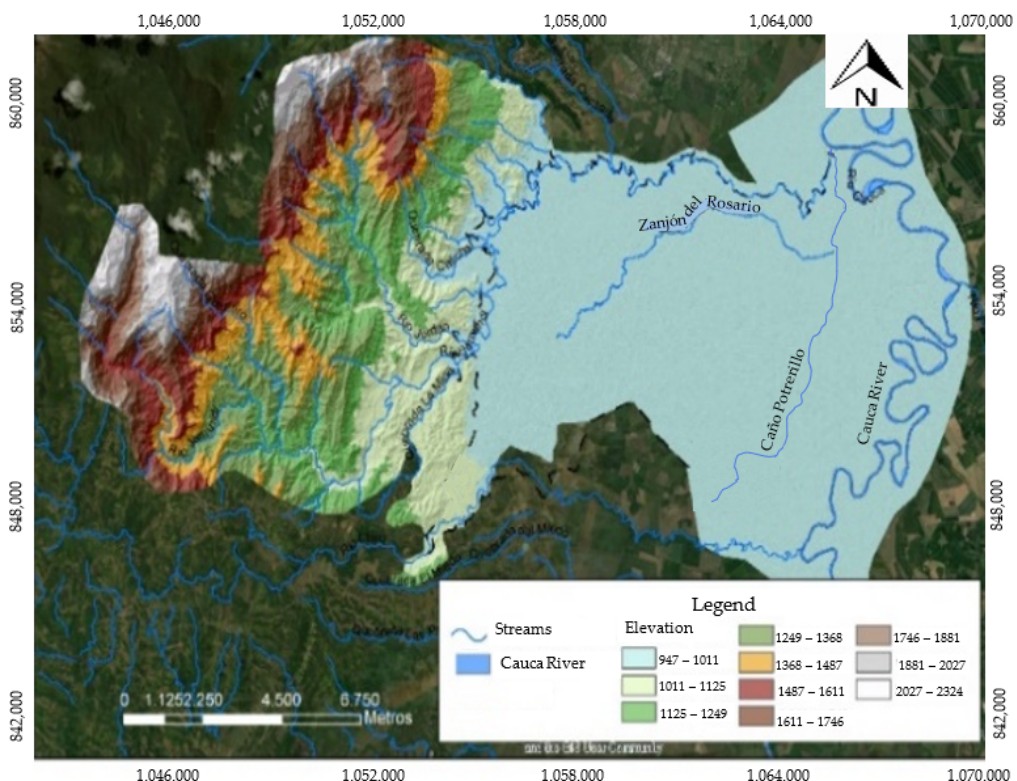

**Figure 4.** Digital elevation model generated at a scale of 1:2000, including the Jamundí River basin. Source: [49].

### 4.2.2. Hydrological Information

The hydrological information consists of the hydrographs of the debris flows corresponding to the return periods defined in the methodology. These hydrographs were obtained by hydrological modeling performed using HEC-HMS software version 4.8. The temporal distribution of multiannual average rainfall recorded at four stations located in the Jamundí River basin, which were used in the hydrologic modeling, is shown in Figure 5. Figure 6 shows the hydrographs of several of the floods introduced at the upstream boundary of the model; this figure also shows the hydrographs presented at the Puente Carretera station during the floods that occurred on 11 November 2011 and 31 December 2019.

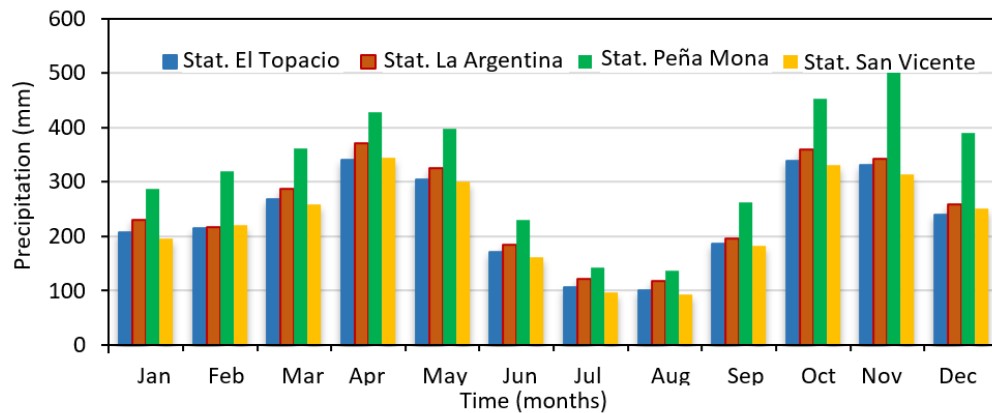

**Figure 5.** Temporal distribution of multiannual average precipitation recorded at four stations located in the Jamundí River basin. Source: [49].

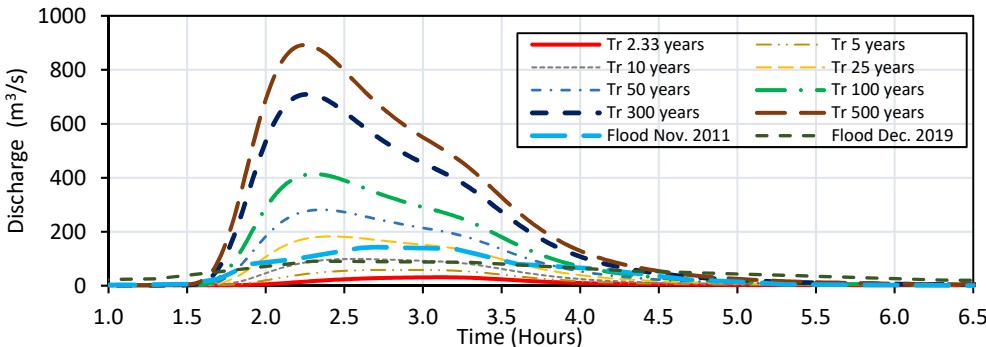

**Figure 6.** Hydrographs of floods introduced in the upstream boundary of the mathematical model of the Jamundí River basin. Source: [49].

### 4.2.3. Sedimentological Information

The sedimentological information corresponded to the estimated volumes of sediments that can be mobilized during debris flow. These volumes can come from the destabilization of slopes adjacent to the channel, from the pluvial erosion of the basin, and from the erosion of the main channel. Table 4 shows the sediment volumes calculated at the upstream boundaries for the event corresponding to the 500-year return period.

**Table 4.** Sediment volumes are introduced into the upstream boundary of the model.

| Return Period (Years) $(m^3/s^2)$ | Sediment Volume Accumulated at the Upstream Boundary $(10^6 \ m^3)$ |
|---|---|
| 2.33 | 0.00 |
| 5 | 0.14 |
| 10 | 0.30 |
| 25 | 0.55 |
| 50 | 0.79 |
| 75 | 0.96 |
| 100 | 1.05 |
| 200 | 1.43 |
| 300 | 1.56 |
| 400 | 1.74 |
| 500 | 1.84 |

The volume of solids resulting from slope erosion, corresponding to laminar erosion, was calculated using the Sediment module of the HEC-HMS model, which applies the Universal Soil Loss Equation (USLE). The volume of solids originating from landslides, rock falls, and flows was calculated by adopting a slope stability model based on the solution

of the equilibrium state in each cell of the domain subjected to one-dimensional rainfall infiltration. The volume of solids arising from slope instability due to lateral channel scour was calculated by comparing the shear stress generated by the flow on the banks of the main channel with the critical shear stress from which particulate material is mobilized from the banks.

### 4.2.4. Rheological Characteristics of the Flow

Based on the characteristic diameters and clay contents found in the Jamundí River basin, it was established that debris flows in this basin have rheological characteristics similar to those of the mudflow of the Colorado Rocky Mountains (USA) near the city of Aspen—pit 1. Consequently, to calculate the values of the yield stress and the absolute viscosity of the flow, the same values of the empirical coefficients of this mudflow were adopted in this modeling; that is, the following values were used: $\alpha1 = 3.60 \times 10^{-2}$ poises, $\alpha2 = 1.81 \times 10^{-1}$ dynes/cm$^2$, $\beta1 = 22.1$, and $\beta2 = 25.7$ [50].

### 4.3. Phase I: Determination of the Hazard in the Rural Area

The first step in determining this hazard is the implementation of a hydrodynamic model that allows analyzing the behavior of debris flows corresponding to different return periods. The modeling was carried out using the FLO2D software version 6.0 since it is a model widely used for modeling this type of event, takes into account the rheology of non-Newtonian flows, allows working with complex topographies, and has the capacity to incorporate sediments from slopes and rockfalls.

The modeling was carried out using a DEM of the study area generated at a scale of 1:25,000. A mesh size of 7 m was adopted, resulting in approximately one million cells. This cell size allowed obtaining adequate accuracy of the results and a reasonable computational time. A Maning roughness value of 0.035 was defined for the Jamundí River, a value of 0.08 for forests and semi-natural areas, and a value of 0.015 for artificial areas.

Debris flow corresponding to the return periods of 2.33, 5, 10, 25, 50, 75, 100, 200, 300, 400, and 500 years were modeled. Each of the simulations took approximately 20 h. The results of these simulations allowed obtaining the variation of the depths and velocities of the flow in each of the pixels through which the territory was represented. Figure 7 shows the maximum depths and velocities obtained for the debris flows with a return period of 500 years.

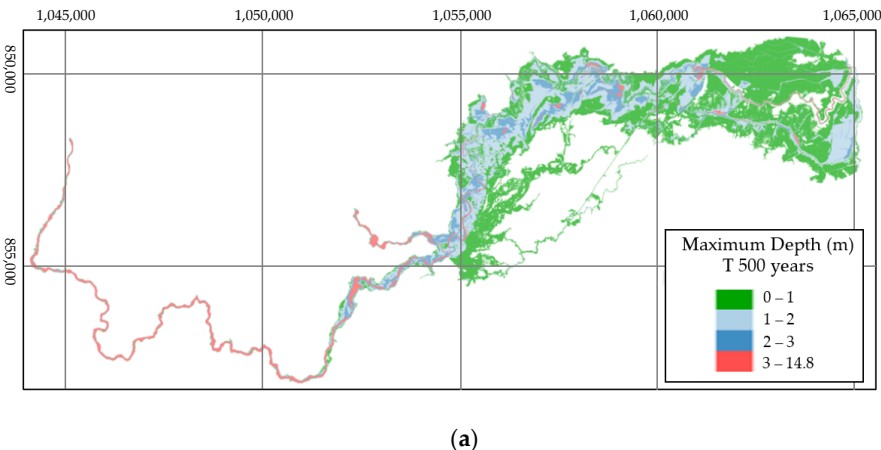

(**a**)

**Figure 7.** *Cont.*

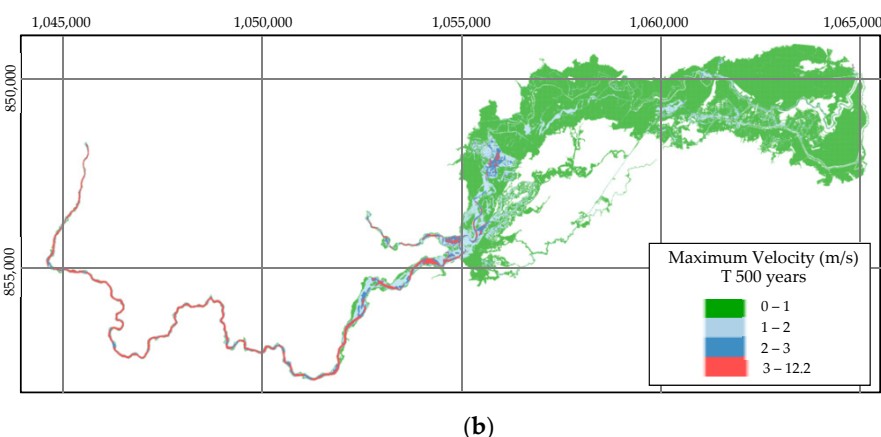

(**b**)

**Figure 7.** Results of the mathematical modeling at 1:25,000 scale of the debris flows with a 500-year return period: (**a**) maximum depths; (**b**) maximum velocities.

Using the results obtained via mathematical modeling and Equation (1), the $I_{DF}$ values were calculated at each pixel for all modeled events. Figure 8a shows the $I_{DF}$ calculated for the event corresponding to the 500-year return period. Subsequently, by means of Equation (2), taking as the $T_{min}$ value the return period from which the flood begins and based on the $I_{DF}$ obtained for each of the events analyzed, the combined $I_{DF}$ value was obtained for each pixel. Figure 8b shows the combined $I_{DF}$ calculated in the Jamundí River basin.

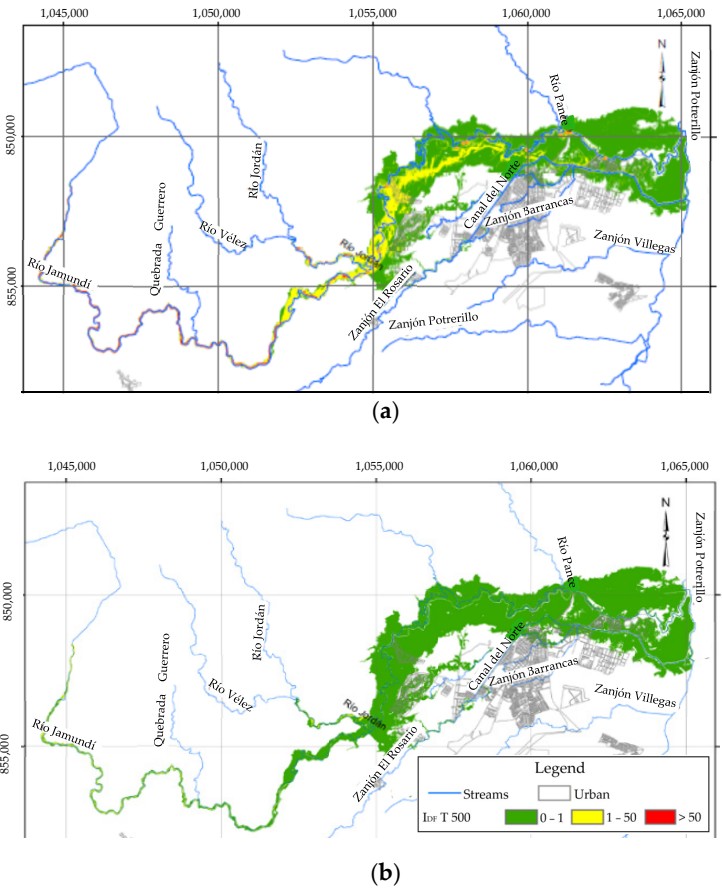

**Figure 8.** Debris Flow Intensity Index, $I_{DF}$, obtained using the results of mathematical modeling at a scale of 1:25,000: (**a**) $I_{DF}$ for the debris flow with a return period of 500 years; (**b**) combined $I_{DF}$.

The next step is to determine the $D_{90}$ value of the sediments that could be mobilized by the debris flows. Figure 9 presents the distribution of the $D_{90}$ diameter of the sediments in the Jamundí River watershed, which was obtained via information collected in field campaigns carried out by [49].

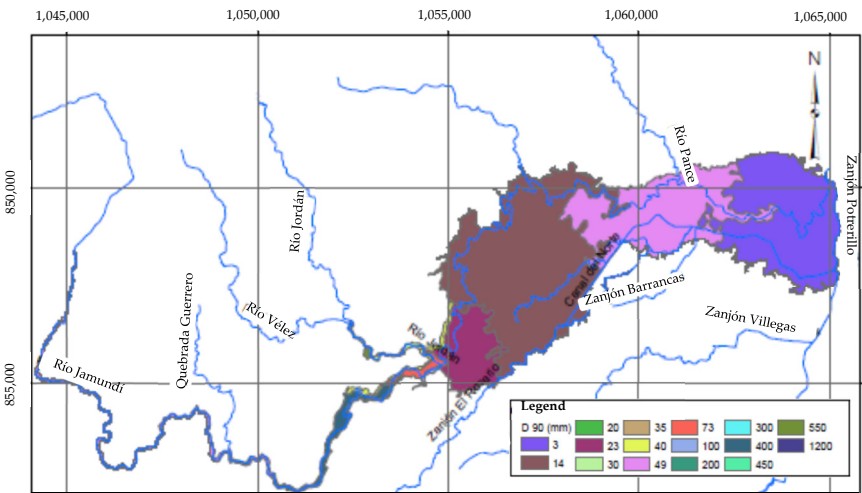

**Figure 9.** Distribution of the $D_{90}$ diameter of the sediments in the Jamundí River watershed. Source: [49].

Finally, to calculate the hazard, the combined $I_{DF}$ and $D_{90}$ values are integrated for each pixel according to the criteria established in Table 1. This procedure allows the classification of the hazard as high, medium, or low in all points of the territory. Figure 10 shows the zoning of the hazard due to debris flows in the Jamundí River watershed at a scale of 1:25,000. According to the results obtained, almost the entire rural area of the watershed presents a low hazard level; only a few small sectors restricted mainly to the watercourses present medium and high hazard levels.

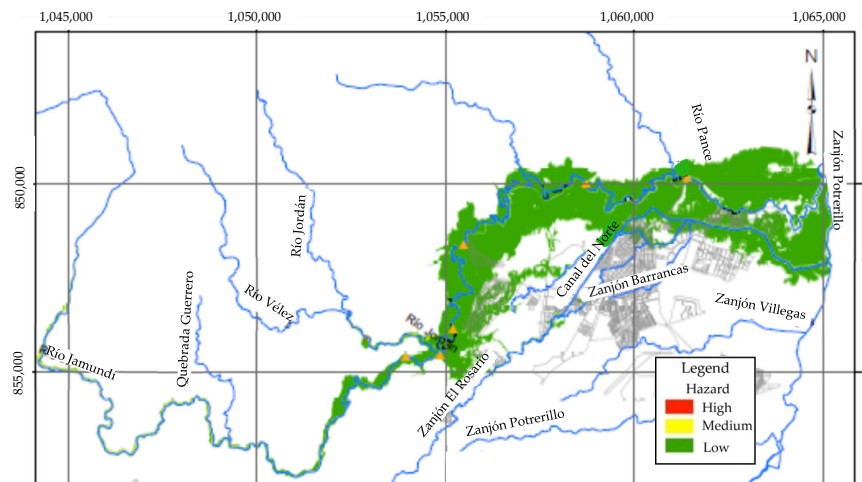

**Figure 10.** Zoning of the hazard at a scale of 1:25,000 due to debris flows in the Jamundí River basin.

*4.4. Phase II: Determination of Hazards in Urban Areas, Urban Expansion Areas, and Areas Classified as High and Medium Hazards in Rural Areas*

Given that the results achieved in the analysis of the rural area indicated that in this sector, the areas exposed to high or medium hazards are very small and mainly limited to the main channels, in this stage of the methodology, the analysis focused on the zoning of the hazard in urban and urban expansion zones, which are located mainly in the lower part of the watershed.

For the mathematical modeling, the Hec Ras model—Debris Flow module was used, which has been successfully used in some modeling of similar events [51] (e.g., Sánchez et al., 2020) and allows the modeling of non-Newtonian flows with a high concentration of solids and calculation of the corresponding internal losses by representing the flow via a single phase.

The modeling was carried out using a DEM at a 1:2000 scale of the lower part of the watershed. In this sector, the slopes are relatively low, so it is mainly considered a deposit area for the material transported by the debris flows where erosion and sediment dragging phenomena will probably be of low magnitude. The same roughness and rheological properties of the flow established in the 1:25,000 scale model were adopted. A cell size of 5 m and a time interval of 2 s was established, which guaranteed the stability of the modeling and allowed the achievement of sufficiently accurate results and reasonable computational times.

Mathematical modeling of eleven events corresponding to the return periods 2.33, 5, 10, 25, 50, 75, 100, 200, 300, 400, and 500 years was performed to establish the hazard curves in each of the flooded pixels. The simulation time for events with the highest return periods was around 14 h. Figure 11 shows the maximum depths and velocities corresponding to the debris flow with a return period of 500 years.

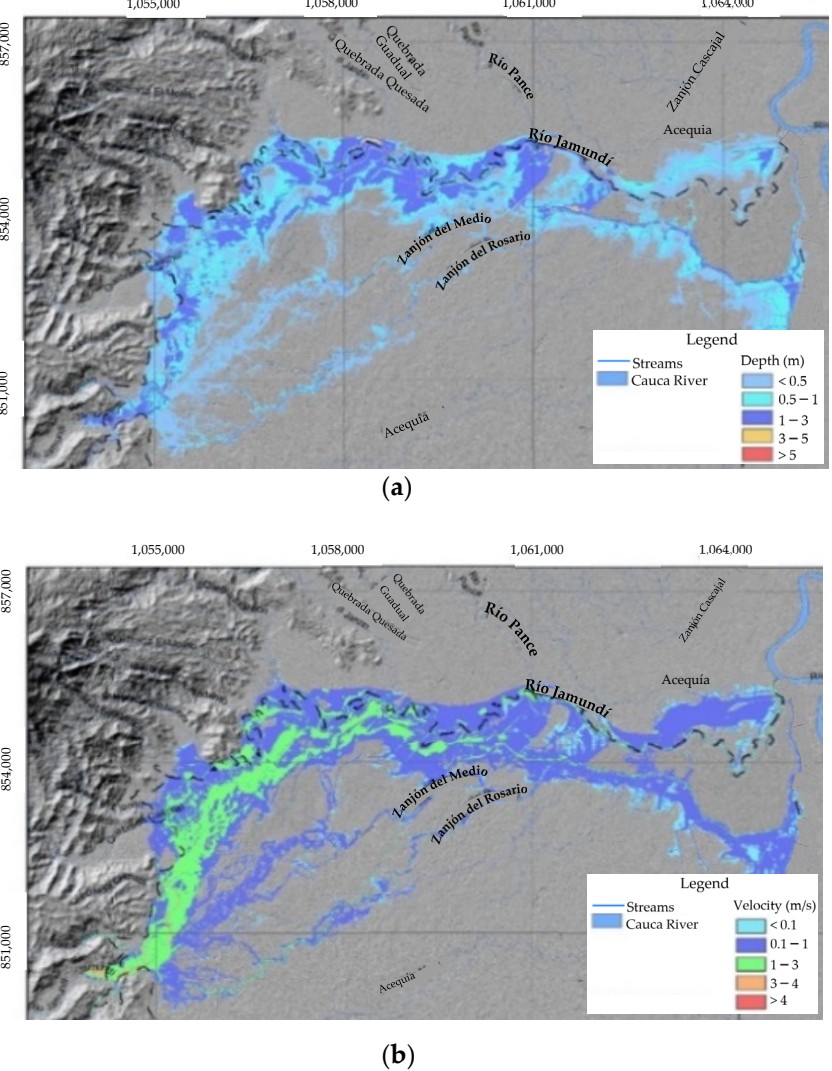

**Figure 11.** Results of the mathematical modeling at a scale of 1:2000 of the debris flow with a return period of 500 years: (**a**) maximum depths; (**b**) maximum velocities.

Based on the results obtained, the $I_{DF}$ in the flooded pixels was calculated. From this information, for each pixel, a hazard curve that relates the probability of occurrence of debris flow with its corresponding $I_{DF}$ was interpolated, taking into account that in each pixel, there are as many $I_{DF}$ as events that flood it. A minimum number of 6 points was established to plot these curves. During the plotting of the curves, it was found that in most cases, the best fit to the points obtained was reached with polynomial equations of degree 2.

In order to carry out the zoning of the territory, the impact of events with a high frequency of occurrence was initially determined. With this objective, based on the hazard curves, the return period that a debris flow with an $I_{DF}$ of 5 m$^3$/s$^2$ would have been established. According to Table 2, if this period is less than 30 years, the hazard is classified as high; if it is between 30 and 100 years, the hazard is medium, and if it is greater than 100 years, the hazard is low. Figure 12a shows the hazard zoning considering this criterion.

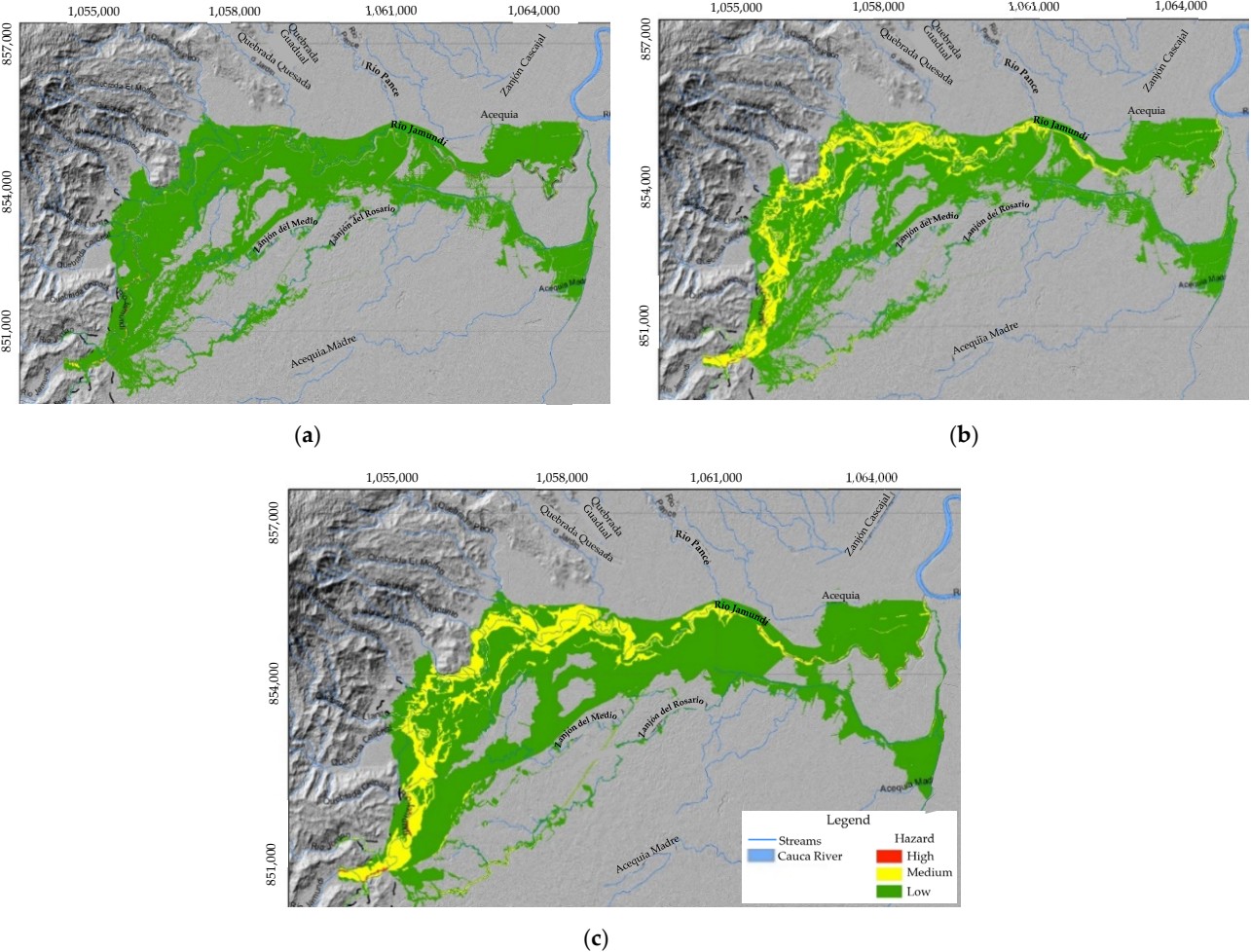

**Figure 12.** Zoning at a scale of 1:2000 of the debris flow hazard in the lower part of the Jamundí River basin: (**a**) zoning obtained from the determination of the return period corresponding to an $I_{DF}$ de 5 m$^3$/s$^2$; (**b**) zoning obtained from the $I_{DF}$ calculation corresponding to an event with a return period of 500 years; (**c**) definitive zoning of the hazard.

Subsequently, the impact of events with a low frequency of occurrence was established, for which the value of the $I_{DF}$ for the event with a return period of 500 years was determined. According to Table 3, if this $I_{DF}$ is less than 1 m$^3$/s$^2$, the hazard is low; if it is between 1 and 25 m$^3$/s$^2$, the hazard is medium, and if it is greater than 25 m$^3$/s$^2$, the hazard is high. Figure 12b shows the zoning of the territory considering this criterion.

The definitive hazard due to debris flows is obtained by aggregating the hazards obtained by considering the impact of frequent events and infrequent events. This aggregation consists of assigning to each pixel the highest level of hazard found. Figure 12c shows the definitive zoning at scale 1:2000 of the hazard due to debris flows in the lower part of the Jamundí River basin. The results indicate that only some sections of the main channel in the upper part of the studied area present a high hazard, which represents a very low percentage of the total affected area. Approximately 25% of the flooded area is subject to a medium hazard level, and the remaining 75% presents a low hazard.

## 5. Discussion

The results obtained by implementing the proposed methodology in the Jamundí River basin made it possible to establish that the areas at high hazard due to debris flows correspond to sectors located near the Jamundí River and several of its tributaries. Some of these sectors are urbanized, and others are used for agricultural purposes, mainly rice and sugar cane crops, which are located to the northwest and east of the study area.

The tributaries around which these high-hazard zones occur correspond to canals that were initially designed and built mainly for irrigation and drainage of sugar cane and rice crops, with relatively low depth, slope, and hydraulic capacity. However, these canals are currently used as elements of the urban rainwater system without having been modified in their geometric and hydraulic characteristics.

In these areas, there is a high probability of debris flows corresponding to return periods of less than 30 years with $I_{DF}$ flow intensity indexes equal to or greater than 5 m$^3$/s$^2$ and the potential to cause partial structural damage to homes and other exposed buildings. There could also be debris flows corresponding to return periods of 500 years or more with flow intensity indexes equal to or greater than 25 m$^3$/s$^2$, which would have the potential to cause severe structural damage.

Within these areas, events with a return period of 25 years would have maximum depths of 5.8 m and maximum velocities of 12.8 m/s. Events with a return period of 500 years would have maximum depths of 7.2 m and maximum velocities of 12.8 m/s.

Some of the areas that present medium hazards are urbanized, while others correspond to crops, pastures, and natural areas. In these areas, debris flows corresponding to a return period of 500 years or more could occur with flow intensity indexes $I_{DF}$ equal to or greater than 1 m$^3$/s$^2$ and less than 25 m$^3$/s$^2$. There could also be events with return periods between 30 years and 100 years that reach $I_{DF}$ values of 5 m$^3$/s$^2$. It is considered that events of these magnitudes put people's lives at risk and forced them to evacuate their homes.

In these areas, events with a return period of 25 years would have maximum depths of 3.4 m and maximum velocities of 3.7 m/s, and events with a return period of 500 years would have maximum depths of 5.8 m and maximum velocities of 5.6 m/s.

Among the areas with low hazards are the urban areas of the municipality of Jamundí, some urban expansion areas, and non-urbanized areas primarily consisting of crops, pastures, and natural spaces. In these areas, debris flows corresponding to return periods greater than 100 years with flow intensity indexes $I_{DF}$ equal to or greater than 5 m$^3$/s$^2$ may occur, which have the potential to cause damage to homes and other exposed buildings. Floods with a return period of 500 years would probably cause slight structural damage and/or some sedimentation since they would have a flow intensity index equal to or less than 1 m$^3$/s$^2$.

In these areas, events with a return period of 25 years would have maximum depths of 2.4 m, and maximum velocities of 1.8 m/s. During events with a return period of 500 years, maximum depths would reach values of 3.2 m and maximum velocities of 2.9 m/s.

In order to control and reduce the volumes of sediments that can potentially reach the riverbeds and generate debris flows, it is advisable to protect and reforest the Jamundí River basin with native species, particularly the areas adjacent to the rivers and streams located in the upper and middle parts of the basin that present erosive processes.

Likewise, the implementation of appropriate land uses in the basin must be promoted in order to reduce rainfall erosion and slope destabilization and increase water retention. This would make it possible to reduce the volumes of water and sediment that will reach the riverbed and, consequently, reduce hazard levels.

In those sectors of the watershed where changes in land use and land cover (deforestation, mining, and extensive cattle ranching) are currently occurring, leading to a decrease in the capacity of water regulation, environmental education activities should be implemented, as well as environmental evaluation, control and monitoring actions for the protection and integral recovery of the watershed.

The assessment of debris flow hazard in the Jamundí River basin demonstrated that the proposed methodology is conceptually clear, grounded in solid theoretical foundations, and straightforward to implement due to its well-defined procedure. Each phase of the methodology provides clear guidance on the steps and their sequence, ensuring that the resulting maps effectively represent the existing hazard levels within a specific area.

Given the serious implications that debris flows could have in urban areas, urban expansion areas, and areas classified as high and medium hazards in rural areas, the information necessary for the hazard assessment in these sectors must have a relatively high level of detail and precision to minimize the uncertainty associated with the results generated by the methodology.

## 6. Conclusions

This paper presents a new methodology for determining the hazard due to debris flows. This methodology takes into account all the relevant technical aspects in the generation of events of these characteristics, which is why it is considered that, from the technical point of view, it is rigorous but, at the same time, simple to implement.

In this methodology, the hazard is classified as high, medium, or low based on the granulometry of the sediments that could be incorporated into the flow and a Debris Flow Intensity Index, $I_{DF}$, which is calculated from the maximum depths and velocities of debris flows corresponding to different return periods. The hydrodynamic characteristics of the flow must be estimated via mathematical modeling performed using models that consider the non-Newtonian flow rheology and the volumes of sediments that could be incorporated into the flow.

Initially, the hazard in the rural area must be calculated using the information at scale 1:25,000 or lower, and later, using the information at scale 1:2000, the hazard must be determined in urban areas and urban projection areas as well as in rural areas classified as a high or medium hazard in the analysis of the rural area.

The hazard classification is based on the reference values adopted for sediment size and for the Debris Flow Intensity Index, $I_{DF}$, which is an empirical parameter. These values were derived from the consequences of several debris flows around the world. In order to increase the level of certainty of the adopted values, it is necessary to expand and permanently update the number of events analyzed for the definition of the $I_{DF}$ and the sediment size. This greater certainty in the adopted reference values would allow an increase in the level of accuracy of the zoning of the territory.

The applicability of the methodology was determined via the implementation of a case study. The results obtained indicate that the areas exposed to a high hazard due to debris flow in the Jamundí River basin are very small, while approximately 85% of the areas that would be affected have a low hazard, and the remaining 15% are exposed to a medium hazard level. Additionally, it is observed that in the Jamundí River basin, the highest levels of hazard are generated by events with a low frequency of occurrence since $I_{DF}$ would reach values that are associated with severe structural damage. Events with a high frequency of occurrence reach $I_{DF}$ values associated with structural damage considered relatively minor.

In order to achieve a comprehensive management of the risk generated by debris flows, it is advisable to integrate the results obtained via the implementation of this methodology with a geomorphological analysis. This analysis would allow us to reconstruct the

fluvio-torrential history of the territory, establish the geoforms indicative of deposits, and identify the geomorphological and morphodynamic elements that favor the occurrence of fluvio-torrential events. This information could be used to establish the geomorphological susceptibility to debris flows, which would allow for refining the hazard classification. Additionally, the results of the geomorphological analysis can be used to calculate more accurately the input information to the mathematical models used to determine the hydro-dynamic characteristics of the debris flows.

The implementation of the methodology proposed in this paper allows us to accurately identify the level of hazard of the territory due to debris flows. This knowledge can be very useful for environmental authorities and organizations in charge of territorial planning and civil protection since it could be used in the design of plans and actions to mitigate the risk generated by events of these characteristics.

**Author Contributions:** Conceptualization, C.A.R., R.A.B. and E.d.J.S.; methodology, R.A.B. and C.A.R.; software, R.A.B. and M.P.L.V.; validation, R.A.B. and M.P.L.V.; formal analysis, R.A.B., C.A.R. and E.d.J.S.; investigation, R.A.B., C.A.R. and E.d.J.S.; resources, C.A.R. and E.d.J.S.; data curation, C.A.R. and E.d.J.S.; writing—original draft preparation, R.A.B.; writing—review and editing, C.A.R., E.d.J.S. and R.A.B.; visualization E.d.J.S.; supervision, E.d.J.S. All authors have read and agreed to the published version of the manuscript.

**Funding:** This research was funded by the Colombian Ministry of Housing, City and Territory, the Colombian Geological Service, the University of Valle via the Convenio Interadministrativo Nº 719 de 2020 and the Convenio Especial de Cooperación No. 033 de 2020, and by the Dirección General de Investigaciones of Universidad Santiago de Cali under call No. 02-2023.

**Data Availability Statement:** Data are available by requirement.

**Acknowledgments:** The authors express their thanks to the Universidad del Valle for providing the information used in this research, to the Colombian Ministry of Housing, City and Territory and the Colombian Geological Service for their technical and financial support, and to the staff of the Office of Risk Management of the municipality of Jamundí for their help in obtaining and organizing the information.

**Conflicts of Interest:** The authors declare no conflict of interest.

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
