# Peer review of "Determination of Hazard Due to Debris Flows"

_water, doi:10.3390/w15234057_

Round 1
Reviewer 1 Report
Comments and Suggestions for Authors
The authors provided a detailed and quality analysis of the debris flow concerning the possible flooding hazard. The literature review elaborates on the experiences and methods for analyzing the analyzed topic.
The authors have justified the application of equation 2 to define the research methodology.
After several readings, I am proposing a minor revision.
-The authors should elaborate and enclose real hydrograms of the analyzed site.
-there is a lack of presentation of the precipitation data in order to fulfill the analysis.
Comments on the Quality of English LanguageSpelling checker and detailed reading of the paper should be done for avoiding the typo, style and grammar errors.
Author Response
Dear Reviewer:
Thank you very much for the review of the article. The comments and questions made were carefully analyzed and discussed by us. The results of this process allowed to improve the quality of the article. In addition, English language was improved. The changes made are highlighted in red. Here are the responses to each of the comments.
GENERAL EVALUATION
|
Questions for General Evaluation |
Reviewer’s Evaluation |
Response and Revisions |
|
Does the introduction provide sufficient background and include all relevant references? |
Yes |
|
|
Are all the cited references relevant to the research? |
Yes |
|
|
Is the research design appropriate? |
Can be improved |
The research design has been improved by implementing the suggestions made by the reviewers |
|
Are the methods adequately described? |
Yes |
|
|
Are the results clearly presented? |
Yes |
|
|
Are the conclusions supported by the results? |
Yes |
|
COMMENTS
Comment 1. The authors should elaborate and enclose real hydrograms of the analyzed site.
Response: In Figure 6 of the new version of the manuscript (Figure 5 in the first version) we include the hydrographs of the floods that occurred on November 11, 2011 and December 31, 2019.
Comment 2. -there is a lack of presentation of the precipitation data in order to fulfill the analysis
Response: In the new version of the manuscript we include a new figure (Figure 5) that presents the temporal distribution of precipitation recorded at four stations located in the Jamundí river basin.

Reviewer 2 Report
Comments and Suggestions for Authors
Dear Editor,
I have carefully read the manuscript by Bocanegra et al. The topic raised in the manuscript is very important, since it concerns dangerous geological phenomena and their impact on people. My experience in debris flow studies is primarily related to marine processes. They differ in many ways from similar phenomena on the onshore, but they also have common features. My comments on the manuscript are based primarily on my experience in the field of marine landslides and debris flows.
When authors write about the intensity of debris flows, they should have mentioned at least briefly the marine analogues. For example, in lines 82-83 authors write that sediment concentrations can vary from 40 to 80%, but in marine environments these values can vary from 25 to 100% (e.g. Shanmugam, 2016, J. Paleogeography).
The authors take granulometric parameters into account in their models, which seems to be a very correct solution. In particular, the authors in lines 166-172 suggest using the D90 parameter. However, the authors did not indicate how exactly this parameter can be obtained. Is this the 90% quantile on the cumulative curve or something else? Authors should provide an unambiguous description of D90.
Lines 212-213 and 137-138 list the various models. I would like to hear the authors' preference and explanation of the reason.
The authors should more clearly define the grain size classification they use. For example, in rows 175-177, the authors refer to a classification where boulders are defined as having a size larger than 50 cm. However, in sedimentology, a different lower limit is usually used for boulders - 25.6 cm. Below, in lines 177-184, the boundaries are already different, and, in particular, a 1 m boundary is used, separating large sediments. In other words, their final classification is different from the one they referred to (source 36).
The coefficients alpha and beta are used in lines 350-351, but nowhere in the text is it explained which formulas use these coefficients.
I also have a few comments about the Figures. For example, the study areas in Figures 2 and 3 are not the same. In Figure 4, the word “anos” is used instead of years. The legend to Figures 6 and 9 is very difficult to read; its size should be increased. Figure 9 covers part of the text.
In general, the ms can be published in a journal after a medium to major revision.
Author Response
Dear Reviewer:
Thank you very much for the review of the article. The comments and questions made were carefully analyzed and discussed by us. The results of this process allowed to improve the quality of the article. In addition, English language was improved. The changes made are highlighted in red. Here are the responses to each of the comments.
GENERAL EVALUATION
|
Questions for General Evaluation |
Reviewer’s Evaluation |
Response and Revisions |
|
Does the introduction provide sufficient background and include all relevant references? |
Can be improved |
Three additional methodologies for determining the hazard due to debris flows were included in the introduction. Considering the two methodologies that had already been described, a total of five different procedures available in the literature for determining this hazard are described in the introduction |
|
Are all the cited references relevant to the research? |
Can be improved |
Some additional references were included to improve the contextualization and argumentation of the article |
|
Is the research design appropriate? |
Yes |
|
|
Are the methods adequately described? |
Can be improved |
In order to improve the description of the methodology, a new figure was included (Figure 2 in the new version of the manuscript) that contains a flow chart in which the procedure that must be followed to implement the proposed methodology is explained in detail |
|
Are the results clearly presented? |
Can be improved |
In the Discussion section, the results obtained were explained in detail and some of the graphs that present these results were improved |
|
Are the conclusions supported by the results? |
Yes |
|
COMMENTS
Comment 1. When authors write about the intensity of debris flows, they should have mentioned at least briefly the marine analogues. For example, in lines 82-83 authors write that sediment concentrations can vary from 40 to 80%, but in marine environments these values can vary from 25 to 100% (e.g. Shanmugam, 2016, J. Paleogeography).
Response: In order to reference marine events, we include in section 2 a paragraph mentioning the type of flow, concentrations and rheology of debris flows in the marine environment.
Comment 2. -The authors take granulometric parameters into account in their models, which seems to be a very correct solution. In particular, the authors in lines 166-172 suggest using the D90 parameter. However, the authors did not indicate how exactly this parameter can be obtained. Is this the 90% quantile on the cumulative curve or something else? Authors should provide an unambiguous description of D90.
Response: D90 is a value obtained from the granulometric curve and represents the diameter such that 90% of the sediment diameters are smaller than this value. In the article we added the following sentence "considering that D90 corresponds to a diameter such that 90% of the sediment diameters are smaller than this value".
Comment 3. Lines 212-213 and 137-138 list the various models. I would like to hear the authors' preference and explanation of the reason.
Response: A paragraph was included in section 3.1 and a sentence in section 3.2 indicating that the selection of the most convenient model is a function of the information available, the type of flow representation and the assumptions and limitations presented by each of these models. In sections 4.3 and 4.4 it was explained that the models used were selected because they have been successfully used in the modeling of events of these characteristics.
Comment 4. The authors should more clearly define the grain size classification they use. For example, in rows 175-177, the authors refer to a classification where boulders are defined as having a size larger than 50 cm. However, in sedimentology, a different lower limit is usually used for boulders - 25.6 cm.
Response: A phrase was included in section 3.1 explaining that there are several classifications, giving the reference in which several of them can be found and indicating the classification selected in the present methodology
Comment 5. Below, in lines 177-184, the boundaries are already different, and, in particular, a 1 m boundary is used, separating large sediments. In other words, their final classification is different from the one they referred to (source 36).
Response: At this point it is not intended to establish a new sediment classification, but rather, considering that some authors have pointed out that a boulder of a size of 1 meter has a great destructive power, the methodology intends to indicate that boulders of a size greater than 1 meter have a more severe impact than smaller boulders, thus generating a higher level of hazard.
Comment 6. The coefficients alpha and beta are used in lines 350-351, but nowhere in the text is it explained which formulas use these coefficients.
Response: These coefficients are used by the mathematical model to calculate the yield stress, τy, and the dynamic viscosity, μ. The equations to calculate these parameters are not presented in the article since they are used in the calculations carried out by the mathematical model internally.
Comment 7. Also have a few comments about the Figures. For example, the study areas in Figures 2 and 3 are not the same. In Figure 4, the word “anos” is used instead of years. The legend to Figures 6 and 9 is very difficult to read; its size should be increased. Figure 9 covers part of the text.
Response: Figure 2 presents the study area for the determination of the torrential flood hazard. Figure 3 presents the DEM constructed for the studies required to determine the hazard; this DEM includes additional perimeter areas to the study area. In order to avoid confusion, we changed the title of Figure 3 to "Digital Elevation Model constructed at a scale of 1:2000 including the Jamundi river basin".
The word años was replaced by years
The size of the legends in Figures 8 and 11 (Figures 6 and 9 in the first version of the manuscript) was increased.
The location of Figure 11 (Figure 9 in the first version of the manuscript) was improved.

Reviewer 3 Report
Comments and Suggestions for Authors
The manuscript focuses on the development and application of a simple methodology for debris flow hazard mapping, potentially exportable to all mountain settings for which some basic information are available.
The methodology described is valuable for publication, but the manuscript needs some improvements.
The introduction needs to be improved with some case studies on debris flow hazard estimation with different methodologies and areas of application in order to provide a complete framework of the context.
Many parts are poorly described. It is not clear whether some parameters were measured, acquired or modeled.
Methodology needs to be better described.
Discussions are completely absent. In my opinion, the advantages and disadvantages of using this methodology need to be stated.
Please see the attached file for more comments.

Only minor editing of English language is needed
Author Response
Dear Reviewer:
Thank you very much for the review of the article. The comments and questions made were carefully analyzed and discussed by us. The results of this process allowed to improve the quality of the article. In addition, English language was improved. The changes made are highlighted in red. Here are the responses to each of the comments.
GENERAL EVALUATION
|
Questions for General Evaluation |
Reviewer’s Evaluation |
Response and Revisions |
|
Does the introduction provide sufficient background and include all relevant references? |
Can be improved |
Three additional methodologies for determining the hazard due to debris flows were included in the introduction. Considering the two methodologies that had already been described, a total of five different procedures available in the literature for determining this hazard are described in the introduction. |
|
Are all the cited references relevant to the research? |
Can be improved |
Some additional references were included to improve the contextualization and argumentation of the article. |
|
Is the research design appropriate? |
Can be improved |
The research design has been improved by implementing the suggestions made by the reviewers |
|
Are the methods adequately described? |
Must be improved |
In order to improve the description of the methodology, a new figure was included (Figure 2 in the new version of the manuscript) that contains a flow chart in which the procedure that must be followed to implement the proposed methodology is explained in detail |
|
Are the results clearly presented? |
Must be improved |
In the Discussion section, the results obtained were explained in detail and some of the graphs that present these results were improved. |
|
Are the conclusions supported by the results? |
Must be improved |
The included discussion provided a more solid basis for the conclusions drawn. |
COMMENTS
Comment 1. The introduction needs to be improved with some case studies on debris flow hazard estimation with different methodologies and areas of application in order to provide a complete framework of the context.
Response: Three additional methodologies for determining the hazard due to debris flows were included in the introduction. Considering the two methodologies that had already been described, a total of five different procedures available in the literature for determining this hazard are described in the introduction. However, these methodologies are clearly less elaborate than those performed in our research
Comment 2. Many parts are poorly described. It is not clear whether some parameters were measured, acquired or modeled.
Response: The description of several sections of the manuscript was improved through the implementation of the suggestions made in the manuscript, which are described later in the responses to the Additional Comments in the Manuscript.
Comment 3. Methodology needs to be better described
Response: In order to improve the description of the methodology, a new figure was included (Figure 2 in the new version of the manuscript) that contains a flow chart in which the procedure that must be followed to implement the proposed methodology is explained in detail.
Comment 4. Discussions are completely absent. In my opinion, the advantages and disadvantages of using this methodology need to be stated.
Response: We include the new section Discussion in the manuscript. This section describes/explains the results obtained and indicates the advantages and disadvantages of the implementation of the proposed methodology.
ADDITIONAL COMMENTS IN THE MANUSCRIPT
Comment 5 Lines 132-133. It is not specified how you associated the return time with the event. Have you done a statistical analysis of past events? Please clarify this point
Response: These return periods refer to the recurrence interval of rainfall events in the basin, which have been recorded through climatological stations. Subsequently, through hydrological modeling, the flows associated with these precipitations are calculated, and through geotechnical analysis, the volumes of sediment that could be mobilized by these events are calculated.
The methodology does not consider statistical analysis of previous events because, in general, this information is not available.
Comment 6 Lines 166-167-. Please check the paper Sepe et al (2023) in which several factors are analyzed and qualitativly associated to landslide hazard
Response:
According to the information reported by Sepe et al (2023), in section 3.1 several factors that have an influence on the intensity of torrential avenues are included
Comment 7 Line 229. for each pixel involved by the flow
Response. This phrase was included in the manuscript
Comment 8 Lines 230 -231. Please can you provide more detail about it? Do you have never obtained insufficient number of idf? for what return period?
Response: The methodology establishes that an IDF should be obtained for each return period and based on these values plot the hazard curve. A low number of points for plotting the hazard curve (e.g., 6 points or less) occurs when a pixel is flooded only by events with a higher return period; in this case, events with return periods additional to those indicated in this methodology should be modeled, in such a way that the number of points available to draw the curve is increased.
Comment 9 Lines 251-260. Sorry but this section is not very clear to me. In my opinion a flow chart could be very useful to follow each of the step of the methodology and to clarify the difference in rural and urban area.
Response: In order to improve the description of the methodology, a new figure was included (Figure 2 in the new version of the manuscript) that contains a flow chart in which the procedure that must be followed to implement the proposed methodology is explained in detail.
Comment 10 Line 285. Geological information could be useful information, also in order to consider the importance of the debris size
Response: Geological information of the study area was included in section 4.1
Comment 11 Lines 315 -316. Were the hydrographs derived from a rainfall input? and was this input set to a particular cell or was a rain-on-grid approach used? Do the return times of the debris flow match those of the rainfall event? If so, please provide more details on the source of the rainfall data, its accuracy, and the methodology used to derive the return time.
Response: The hydrographs were obtained by mathematical modeling from the precipitation events recorded in the basin. As is generally the case, we do not have enough information of debris flows that have occurred in the basin, so it is not possible to develop a statistical analysis of this information.
Comment 12 Lines 342-343. Please provide more information about this calculation. Is it a result of numerical modeling (Table 4)?
Response. In section 4.2.3 a paragraph was included describing the procedure implemented for calculating the volumes of sediments introduced in the modeling
Comment 13 Line 346. Is this a generic non-Newtonian flow or does it correspond to a particular type of rheology? On what basis did you establish the similarity between the two debris flow (e.g. final volume, velocity, shape and size of the basin)? please provide more information
Response: At this point we are referring to a generic non-Newtonian fluid. The similarity of this flow with the mudflow of the Colorado Rocky Mountains (USA) was established through the comparison of the characteristic diameters and clay contents of this last flow with the values of these parameters obtained in the Jamundí River basin through of the measurements made by (43). A sentence giving this information was included in section 4.2.2.
Comment 14 Line 433 - 435. This estimation (D90) is also derived from bumerical simulation? please specify it
Response The Distribution of the D90 Diameter of the sediments in the Jamundí river watershed was obtained through data collected in field campaigns carried out by the Universidad del Valle and the Colombian Geological Service. A sentence giving this information was included in section 4.3.
Comment 15 Line 494. This line is not readable
Response: The location of Figure 11 (Figure 9 in the first version of the manuscript) was improved.
Comment 16 Line 529. (Figure 10) this is very similar to the example in figure 1, I suggest eliminating this duplication
Response: This figure was removed

Round 2
Reviewer 2 Report
Comments and Suggestions for Authors
Dear Editor, I have read the authors' responses to my comments and these responses have satisfied me. I believe that the ms can now be published.
Author Response
Dear Reviewer:
Thank you very much for the review of the article.
Reviewer 3 Report
Comments and Suggestions for Authors
The article has been revised based on the comments provided. However, the new Figure 5, which depicts precipitation trends from January to December, is not entirely clear because it is not specified in the figure or even in the text whether this rainfall is an average of a relatively long time series or refers to a specific year. Please improve this aspect.
Author Response
Dear Reviewer:
Thank you very much for the review of the article. We improved the aspect suggested in the review. The changes made are highlighted with yellow shading. Here is the response to the comment.
Comment. The new Figure 5, which depicts precipitation trends from January to December, is not entirely clear because it is not specified in the figure or even in the text whether this rainfall is an average of a relatively long time series or refers to a specific year. Please improve this aspect.
Response: The precipitation values presented correspond to average values of precipitation recorded at stations located in the Jamundí River basin since approximately the beginning of the 1970`s. To clarify this, in the text and in Figure 5 we changed the phrase “temporal distribution of rainfall” by the phrase “temporal distribution of multiannual average rainfall”.
